# Acidosis-Induced TGF-β2 Production Promotes Lipid Droplet Formation in Dendritic Cells and Alters Their Potential to Support Anti-Mesothelioma T Cell Response

**DOI:** 10.3390/cancers12051284

**Published:** 2020-05-19

**Authors:** Natalia Trempolec, Charline Degavre, Bastien Doix, Davide Brusa, Cyril Corbet, Olivier Feron

**Affiliations:** 1Pole of Pharmacology and Therapeutics (FATH), Institut de Recherche Expérimentale et Clinique (IREC), UCLouvain, B-1200 Brussels, Belgium; natalia.trempolec@uclouvain.be (N.T.); charline.degavre@student.uclouvain.be (C.D.); bastien.doix@gmail.com (B.D.); cyril.corbet@uclouvain.be (C.C.); 2Institut de Recherche Expérimentale et Clinique (IREC) Flow Cytometry Platform, UCLouvain, B-1200 Brussels, Belgium; davide.brusa@uclouvain.be

**Keywords:** malignant mesothelioma, acidosis, transforming growth factor-β2 (TGF-β2), dendritic cell, vaccination, metabolism, lipid droplets, diacylglycerol O-acyltransferase (DGAT), C-C chemokine receptor type 7 (CCR7)

## Abstract

For poorly immunogenic tumors such as mesothelioma there is an imperious need to understand why antigen-presenting cells such as dendritic cells (DCs) are not prone to supporting the anticancer T cell response. The tumor microenvironment (TME) is thought to be a major contributor to this DC dysfunction. We have reported that the acidic TME component promotes lipid droplet (LD) formation together with epithelial-to-mesenchymal transition in cancer cells through autocrine transforming growth factor-β2 (TGF-β2) signaling. Since TGF-β is also a master regulator of immune tolerance, we have here examined whether acidosis can impede immunostimulatory DC activity. We have found that exposure of mesothelioma cells to acidosis promotes TGF-β2 secretion, which in turn leads to LD accumulation and profound metabolic rewiring in DCs. We have further documented how DCs exposed to the mesothelioma acidic milieu make the anticancer vaccine less efficient in vivo, with a reduced extent of both DC migratory potential and T cell activation. Interestingly, inhibition of TGF-β2 signaling and diacylglycerol O-acyltransferase (DGAT), the last enzyme involved in triglyceride synthesis, led to a significant restoration of DC activity and anticancer immune response. In conclusion, our study has identified that acidic mesothelioma milieu drives DC dysfunction and altered T cell response through pharmacologically reversible TGF-β2-dependent mechanisms.

## 1. Introduction

Malignant mesothelioma (MM) is an aggressive tumor with long latency period between asbestos exposure and the first sets of symptoms. Treatments include antifolate and platinum-based chemotherapy combined with anti-angiogenic drugs, surgery, and radiotherapy, but combined they only offer on average a gain of one year in overall survival [1]. The recent introduction of immunotherapy in the clinics, in particular immune check-point blockers (ICBs), has therefore brought new hope for MM patients. The outcome of a major clinical trial investigating anti-CTLA4 tremelimumab was however negative [2], and only partial responses have been reported so far with anti-PD1 and anti-PDL1 antibodies [3,4,5,6,7]. Retrospective analyses of available mouse and patient data pinpoint likely explanations for the limited efficacy of ICBs in MM, including a low mutational load [8,9], an immunosuppressive environment [10,11], and a low number of functional infiltrating CD8^+^ T cells [12].

These observations underline the need for a better understanding of the determinants of cold tumors such as MM. In particular, an important area of investigation is how to reveal the naturally occurring source of tumor-associated antigens (TAAs). There is a growing appreciation that TAAs may be released through the cytotoxic effects of radiotherapy, chemotherapy, and probably also targeted therapies [13,14]. The question can however not be restricted to TAA unmasking since a consecutive efficient presentation remains the critical step for taking advantage of spontaneous anticancer T cell activity [15]. Dendritic cells (DCs) have been historically recognized to fulfill this role, and numerous studies have led to a thorough understanding of the mechanisms driving the uptake, processing, and presentation of TAAs [16]. This knowledge actually led to the development of DC-based vaccines to deliver specific antigens to boost anticancer immune response [17]. In MM, several small clinical trials have already explored this possibility using various models of TAA-pulsed DCs [18,19,20,21,22]. More generally, clinical successes reported with DC vaccination have reinforced the view that endogenous DCs encounter some obstacles that prevent them from performing their tasks. 

The tumor microenvironment (TME) is a major contributor to DC dysfunction in cancer, with several tumor and stroma cells-secreted factors identified as negative regulators of DC functionality [23,24]. In addition, physico-chemical features of TME such as hypoxia and acidosis were shown to directly modulate the immune response. While the hypoxia-inducible factor-1α (HIF-1α)-driven gene program directly accounts for functional alterations in immune cell functions under hypoxic conditions [25], acidosis has mainly been linked to a defect in immune response because of associated lactate accumulation in tumors [26]. The release of excess lactate/H^+^ by cancer cells into the extracellular medium actually makes the concentration gradients of these waste products less favorable for T cells to maintain glycolysis [27,28], thereby directly impacting on interferon gamma (IFN-γ) and cytokine production [29,30]. The field of immunometabolism has emerged from such studies where altered bioenergetic and biosynthetic preferences of immune cells were found to be directly interconnected with TME and to influence their functions. In DCs, metabolic disorders were reported to alter their capacity to mature, migrate, and instruct T cells [31]. In particular, accumulation of lipid bodies or droplets (LDs) was consistently documented in the cytosol of tumor-homing DCs, and these LDs were further identified to impair DC activity, including through a defect in antigen presentation and T cell activation [32,33,34]. Although local tumor acidosis was not identified as a trigger of detrimental fatty acid (FA) accumulation in DC, we recently reported that in cancer cells, acidosis could lead to alterations in lipid metabolism [35] and consecutive LD accumulation in a TGF-β2-dependent manner [36]. These data led us to postulate that acidosis could participate to the immunosuppressive tumor environment via the preferential activation of TGF-β2 and its capacity to induce LD accumulation directly into DCs. We confirmed this hypothesis by documenting that exposure of mesothelioma cells to acidosis promotes TGF-β2 secretion, which in turn promotes LD accumulation in DCs. The consecutive metabolic rewiring characterized by a reduced glycolytic flux and respiration was directly associated with the failure of DCs to support T cell activation. We further provided evidence that DCs exposed to the mesothelioma acidic milieu make anticancer vaccines less efficient in vivo, with reduced T cell activation and DC migratory potential. 

## 2. Results

### 2.1. DCs Accumulate Lipid Droplets in Response to TGF-β2 Released by Acid-Adapted Mesothelioma Cells

We have recently reported that tumor acidosis promotes autocrine TGF-β2 signaling in cancer cells, which in turn supports epithelial-to-mesenchymal transition (EMT) and lipid droplet (LD) accumulation [36]. Since TGF-β is a well-known immunosuppressive cytokine, we here examined whether the acidic tumor microenvironment could contribute to immunosuppression in mesothelioma through TGF-β2-induced LD accumulation in dendritic cells (DCs). For this purpose, we developed a model where mesothelioma cancer cells Ab1 and AE17 were chronically adapted to pH 6.5 to mimic in vivo acidosis as previously reported [35,36,37]. We first confirmed a net increase in TGF-β2 release in the extracellular medium of acid-adapted mesothelioma cells vs. parental cells maintained at pH 7.4 (Figure 1A,B). We next examined whether TGF-β2-containing conditioned medium (CM) from pH 6.5-adapted mesothelioma cells (6.5/CM here below) could induce the formation of LDs in dendritic cells (DCs). Treatment of bone marrow-derived DCs (BMDCs) with 6.5/CM led to an increase in intracellular LDs detected by both Oil Red O (ORO) and BODIPY 495/503, vs. CM from mesothelioma cells maintained at pH 7.4 (7.4/CM) and non-conditioned medium (NCM) (Figure 1C,D). This increase in LD content was prevented by SB-431542, a potent TGF-beta type I receptor inhibitor (Figure 1C,D). In addition, we showed that acute treatment of DCs with recombinant TGF-β2 led to an increase in LD formation (Figure 1E). Of note, exposure of BMDCs to control medium buffered at pH 6.5 failed to induce changes in LD content (Appendix A) and the exposure to pH 6.5-buffered medium *per se* only marginally influenced DC survival (Appendix A). 

### 2.2. TGF-β2-Dependent LD Accumulation in DCs Led to Metabolic Reprogramming

We next examined the determinants of FA accumulation within LDs in 6.5/CM-exposed DCs using a medium deprived of lipids and inhibitors of diacylglycerol O-acyltransferase (DGAT), the enzyme involved in the last step of triacylglycerol synthesis. We found that upon exposure to 6.5/CM in the presence of delipidated serum, a net reduction in LD formation was observed (Figure 2A). Although we cannot formally exclude a contribution of FA synthesis to LD formation, these data indicate that accumulation of LDs by DCs was largely dependent on the uptake of exogenous lipids. Inhibition of DGAT1 and DGAT2 enzymes by A922500 and PF-06424439, respectively, led to a dramatic reduction in LD formation in 6.5/CM-exposed DCs (Figure 2B,C). Of note, while both DGAT1 and DGAT2 inhibition inhibited 6.5/CM-induced LD formation, only DGAT2 inhibition reduced basal amounts of LDs (i.e., in the 7.4/CM condition) (Figure 2B,C). We also found that in 6.5/CM-exposed DCs, DGAT2 inhibition more extensively induced cell death than DGAT1 inhibition (Appendix A). While atglistatin (ATGLi), an inhibitor of adipose triglyceride lipase (ATGL), led to a dramatic increase in LD formation in DCs exposed to 7.4/CM, it only marginally influenced the extent of LDs in 6.5/CM-exposed DCs (Appendix A), suggesting that in these cells FA turnover in LDs was not overly stimulated. 

Since LD accumulation under acidosis infers that lipid metabolism is altered in DC, we next examined the status of other major metabolic pathways in DCs exposed to 6.5/CM. Using Seahorse technology, we first measured the extracellular acidification rate (ECAR) as a surrogate for glycolysis, a key pathway known to support Toll-like receptor (TLR)-induced DC maturation and activation [38]. We found a net decrease in ECAR in DCs upon exposure to 6.5/CM (Figure 2D); this effect was TGF-β-dependent since it could be reversed by SB-431542 (Figure 2D). We also showed that acute treatment of DCs with recombinant TGF-β2 led to a significant reduction in both glucose consumption and lactate release (Appendix A). Both DGAT inhibitors also partly rescued ECAR in DCs exposed to 6.5/CM (Figure 2E). We next evaluated the influence of 6.5/CM on DC respiration by measuring oxygen consumption rate (OCR). We found a net decrease in OCR in DCs exposed to 6.5/CM (i.e., to a larger extent than with 7.4/CM) (Figure 2F). TGF-β receptor inhibitor SB-431542 largely reversed the phenomenon (Figure 2F). Part of OCR inhibition was however also directly dependent on the acidic pH (i.e., CM-independent) (Appendix A). Interestingly, OCR was rescued in 6.5/CM-exposed DCs treated with both DGAT1i and DGAT2i (Figure 2G).

### 2.3. Acid-Exposed Mesothelioma Cells Alter the Migratory Capacity of DC

Glycolysis has been documented to be critical to sustain the DC migration and CCR7 oligomerization that is required to promote trafficking towards draining lymph nodes (LNs) [39]. Altered glucose metabolism in 6.5/CM-exposed DCs led us to explore a potential defect in DC trafficking. We found that DC exposure to 6.5/CM led to a significant decline in surface CCR7 expression, which was reversed by a TGF-β signaling inhibitor (Figure 3A and Appendix A). We next evaluated whether metabolic reprogramming using DGAT inhibitors, as shown in Figure 2E, could influence surface CCR7 expression. We found that at the concentrations of the pharmacological inhibitors used, inhibition of DGAT1 activity reversed CCR7 decline, while only a trend was observed with a DGAT2i (Figure 3B). Moreover, reduced CCR7 surface expression correlated with a decreased, TGF-β2-dependent chemotactic migration of CD11c^+^ cells, determined in a transwell assay (Figure 3C). In order to evaluate the effects of tumor acidosis on DC migration and lymph node homing in a more physiological context, we injected labeled DCs (VivoTrack 680 NIR Fluorescent Imaging Agent) into mouse footpad and tracked their migration. After 3 days, popliteal lymph nodes (PLN) were removed and evaluated for presence of migrated DCs (Figure 3D). A significant decrease in the in vivo migratory potential was observed for DCs pre-exposed to 6.5/CM and importantly, inhibition of TGF-β signaling reversed this phenotype (Figure 3D).

### 2.4. CD8^+^ T Cell Response Is Altered upon DC Exposure to the Secretome of Acid-Exposed Mesothelioma Cells

We next aimed to further study whether 6.5/CM exposure could also alter intrinsic DC activity in an optimized antigen recognition model using DCs loaded with ovalbumin (OVA) peptide SIINFEKL and CD8^+^ T cells isolated from OT-1 transgenic mice. Pre-treatment of DCs with 6.5/CM resulted in reduced surface expression of CD40 upon lipopolysaccharide (LPS) administration (Figure 4A). A decrease in the expression of this co-stimulatory molecule was however not reversed by SB-431542 (Figure 4A), nor by DGAT inhibition (Appendix A). Still, 6.5/CM accounted for a more significant reduction in the MHCII^+^ CD40^+^ DC population than that observed upon exposure to pH 6.5-buffered medium (Appendix A). Similarly, secretion of pro-inflammatory interleukin-12 (IL-12p70) was significantly decreased in response to 6.5/CM exposure but not by pH 6.5-buffered medium (Figure 4B and Appendix A). CD80 and CD86 expressions were not or slightly affected by 6.5/CM, respectively, (Appendix A) and were not influenced by SB-431542 or DGATi (Appendix A). 

We finally evaluated the capacity of 6.5/CM-exposed DCs to influence the T cell phenotype. We found that 6.5/CM significantly reduced clonal OVA-specific CD8^+^ expansion in a SB-431542-dependent manner (Figure 4C and Appendix A), an observation confirmed by the reduction in OVA-specific CD8^+^ T cell proliferation upon DC treatment with TGF-β2 (Appendix A). A small but significant reduction in the capacity of 6.5/CM-exposed DCs to induce interferon gamma (IFN-γ) production in activated CD69^+^ OVA-specific CD8^+^ T cells was also detected (Figure 4D and Appendix A). 

### 2.5. Acid-Exposed Mesothelioma Cells Alter the Ability of DC-Based Vaccination to Induce Anti-Tumor Immune Response In Vivo

To further validate the effect of tumor acidosis on DC functionality in vivo, we applied a model of autologous vaccination in mice bearing peritoneal malignant mesothelioma [40]. DC vaccines were generated upon exposure to freeze/thaw-killed mesothelioma cells in the presence of 6.5/CM with or without SB-431542. In a pilot study, mice received one single dose of autologous DC vaccine at day 13 post-mesothelioma cell injection (see Appendix A). While tumor control was obtained in the control DC-vaccinated group, 6.5/CM-exposed DC vaccination failed to lead to inhibition of tumor growth except when the inhibitor of TGF-β signaling was present at the time of DC priming (Appendix A). The large tumor sizes in this experiment prevented further follow-up for ethical issues, so that we next used a protocol where mice received two doses of autologous DC vaccine at days 7 and 14 post-mesothelioma cell injection (Figure 5A). Mice vaccinated with 6.5/CM-exposed DCs showed a reduction in progression-free survival (*p* = 0.057, *n* = 6) (Figure 5B), with relapse occurring in half of the mice post-vaccination (Figure 5C). Importantly, when DCs were exposed to 6.5/CM in the presence of SB-431542, the efficacy of anticancer vaccination was fully restored (Figure 5B,C).

To characterize the lesser efficacy of therapeutic vaccination using 6.5/CM-based DCs, we next evaluated CD8^+^ response using splenocytes isolated from vaccinated mice and in vitro re-challenged with mesothelioma cell lysates. Net decreases in both antigen-specific proliferation (Figure 6A and Appendix A) and IFN-γ production (Figure 6B and Appendix A) were detected 20 days after vaccination. We also found that 6.5/CM-based DC vaccination was associated with a decreased CD8^+^ cytotoxic activity, as reflected by decreased lysosomal-associated membrane protein-1 (LAMP-1) expression in CD8^+^ splenocytes (Figure 6C and Appendix A). The reduction in each of the above parameters (i.e., proliferation, IFN-γ production and LAMP-1 expression) was consistently prevented in CD8^+^ splenocytes collected from mice vaccinated with DCs exposed to 6.5/CM in the presence of the TGF-β inhibitor SB-431542 (Figure 6A–C). Evaluation of CD4^+^ splenocyte response to vaccination with 6.5/CM-based DCs revealed a limited reduction in proliferation and no changes in OX-40 and CD25 surface expression (Appendix A). Reduced IFN-γ production upon phorbol myristate acetate (PMA)/ionomycin stimulation was however observed in circulating CD4^+^ T cells collected from mice vaccinated with 6.5/CM-based DCs (Figure 6D); this effect on IFN-γ^+^ CD4^+^ cells was further dampened in vaccination conditions including both 6.5/CM and TGF-β inhibition (Figure 6D).

## 3. Discussion

Host immune evasion is a hallmark of tumor metastatic progression and treatment resistance. The tumor microenvironment (TME) largely contributes to the failure in developing an efficient anticancer immune response. Beside the presence of cells endowed with immunosuppressive activity in tumors, hypoxia but also acidosis are well-known TME characteristics that prevent the anticancer activity of various immune cells [26,41]. We recently reported that adaptation of cancer cells that face acidosis is ensured by autocrine TGF-β2 signaling that supports an increase in LD accumulation [36]. This finding echoes two independent sets of data in the immuno-oncology field, which are the historical identification of TGF-β as a master regulator of immune tolerance [42], and the demonstration that LD accumulation in dendritic cells (DCs) alters their function [32,33,34,43]. In the current study, we now provide evidence that the acidic tumor milieu promotes LD accumulation by DCs in response to secreted TGF-β2, leading to a profound metabolic rewiring that alters their migratory potential to lymph nodes and impedes their immunostimulatory activity. Importantly, we also document that acid-driven DC dysfunction may be overcome with drugs preventing TGF-β signaling but also blocking LD formation. 

While already reported in the original paper describing DCs [44] and repeatedly reported since then [45], the roles of LDs in DCs were mainly decrypted in the last decade. Bougneres and colleagues initially reported that GTPase Igtp expressed on LD membranes played a critical role in the cross-presentation of phagocytosed antigens to CD8^+^ T cells [46]. Later on, Herber and colleagues reported that many tumor-associated DCs in mice and cancer patients had increased levels of triglycerides stored into LDs, preventing them from processing antigens and thus stimulating T cells [32]. This finding was expanded by others who documented that lipid peroxidation in DC induced activation of the endoplasmic reticulum (ER) stress response factor XBP1, in turn leading to triglyceride accumulation and subsequent blunting of DC activity [34]. In parallel, the group of Gabrilovic provided evidence that oxidatively truncated lipids accumulated into LDs could covalently bind to chaperone hsp 70 and thereby impede peptide–MHC class I (pMHC) complex translocation from the phagosome/lysosome compartments to cell surface [33,43]. Today, the consensual view is that DCs have a propensity to accumulate LDs but in tumors, the excessive uptake and consecutive oxidation of fatty acids lead to alterations in the DC capacity to support T cell-based anticancer immunity. Triggering actors of lipid-evoked defects in DCs, however, remained elusive. Our data have now identified TGF-β, and in particular the TGF-β2 isoform, as a direct link between acidic TME and DC dysfunction. Using a model of mesothelioma cells exposed to acidosis, we have indeed shown that an increase in the secretion of active TGF-β2 drives a significant increase in LD amounts within DCs and alters their metabolic status in such a way that it contributes to a defect in DC function and in particular DC trafficking.

Several reports have documented that DC activation is associated with an increase in glycolytic turnover [31,38]. The preferred metabolic routes for glucose in activated DCs however appear to be double. In some studies, aerobic glycolysis (i.e., glucose to lactate) was actually documented to support the pentose phosphate pathway to generate NADPH as reductive equivalent used to sustain FA synthesis (FAS) [47]. In these Warburg-like conditions, pyruvate is converted into lactate to regenerate the NAD^+^ that is needed to support the increased glycolytic turnover. The other reported fate for glucose-derived pyruvate is to participate to the mitochondrial production of citrate that is then transported to the cytosol to generate acetyl-CoA as a lipogenic building block [48]. FAs are indeed particularly needed for phospholipid production and ER and Golgi apparatus expansion to support increased protein synthesis associated with the DC activation stage [24,31,48]. In our experiments, we found that DCs exposed to mesothelioma acidic milieu exhibited a decrease in both EACR (i.e., glucose to lactate + H^+^) and OCR that somehow reflects a lesser TCA cycle activity. Importantly, both alterations were reversed in the presence of SB-431542, indicating a role of TGF-β in this defective DC activation process. Yet more interestingly, we found that by preventing LD formation using DGAT inhibitors, we could restore the use of glucose in activated DC. The most likely explanation is that by preventing triglyceride formation, DGAT inhibitors render FA more available for critical DC functions and spare pyruvate that may be used for needed tasks (Figure 7). The preferential fate of FA in TGF β-exposed DCs therefore shifts from an excessive accumulation (as triglycerides) within LDs to a readily available pool of lipids for synthesis of membrane phospholipids and β-oxidation. Such a model of preferred FA storage without the capacity to use them is further reinforced by the demonstration that Atglistatin, an inhibitor of TG lipase, led to a dramatic LD accumulation in DCs exposed to 7.4/CM but barely influenced the extent of LDs in 6.5/CM-exposed, suggesting a deficit in basal FA mobilization in these cells. Among the phenotypical traits associated with metabolic rewiring of activated DCs is the increased CCR7-based migratory potential [49]. In our study, we provide evidence that a reduction in both the CCR7 expression and the ability of DCs to migrate towards the draining lymph nodes may be triggered by the tumor acidic environment. Importantly, we further showed that this defect can be reversed (1) by blocking the critical driver of these metabolic alterations, namely TGF-β but also (2) by disrupting the preferred FA fate in this condition, that is the formation of LDs (see Figure 3). Of note, the preferential reversal of CCR7 expression upon DGAT1 inhibition (vs. DGAT2 blockade) coincides with the role of DGAT1 in protecting the ER from the toxicity of excess uptake of exogenous fatty acids while DGAT2 is more prone to mediating TG synthesis of *de novo* synthesized fatty acids [50,51].

As for the defect in DC migration, the incapacity of DCs exposed to acidic CM to stimulate CD8^+^ T cell proliferation was sensitive to TGF-β inhibition (Figure 4C). The latter remarkably translated into the in vivo rescue of DC-based autologous vaccination, including an increased systemic immunity (i.e., IFN-γ^+^ LAMP1^+^ CD8^+^ and IFN-γ^+^ CD4^+^ cell populations). Of note, a third level of regulation of DC activity was observed in response to acidic CM that was strictly due to the acidic pH but independent of LD formation and TGF-β signaling (i.e., DGAT inhibitor- and SB-431542-insensitive). Those acid-induced deficiencies in IL12 secretion and CD40 expression are actually related to an altered CD4^+^ response. Indeed, while IL-12 is involved in the differentiation of naive T cells into Th1 cells [52], CD40 interaction with CD40L of CD4^+^ T cells is known to license DCs to cross-prime naïve T cells [53]. Of note, co-stimulatory molecules CD80 and CD86 remained largely insensitive to changes in pH or exposure to inhibitors of DGAT and TGF-β signaling, in accordance with previous observations by Herber and colleagues [32].

Although most of the data reported in this study may potentially apply to all cancer types, this very study is about mesothelioma. As detailed in the introduction, while ICBs have the potential to reverse T cell exhaustion within MM tumors, combined rescuing of DC function may be of great help to support needed T cell priming and activation [54]. Several clinical studies have actually implicated DC dysfunction in the pathogenesis of mesothelioma, with MM patients exhibiting reduced numbers of circulating DCs and decreased abilities to process antigen and activate T cell response [55]. Increased lipid accumulation and reduced antigen processing ability were actually reported in mesothelioma-infiltrating DCs [56]. Our findings therefore open concrete perspectives to improve DC functions (as adjuvant modalities to ICBs) by either blocking TGF-β2 or DGAT. Interestingly, while DGAT inhibitors are close to being available on the market in the context of obesity and diabetes [57], trabedersen, a specific antisense oligonucleotide directed against the TGF-β2 gene, is already available and currently used in the clinic [58,59,60]. These approaches will have the double advantage of boosting the immune response while also preventing disease progression by blocking metastases, as we recently reported [36]. 

## 4. Materials and Methods

### 4.1. Mesothelioma Cell Culture and Treatments

AB1 and AE17 mesothelioma cell lines were acquired in the last 3 years from the European Collection of Authenticated Cell Cultures (ECACC), where they were regularly authenticated by short tandem repeat profiling. Cells were stored according to the supplier’s instructions and used within 6 months after resuscitation of frozen aliquots. AB1 and AE17 cells were maintained in DMEM (Thermo Scientific, Waltham, MA, USA) supplemented with 10% fetal bovine serum (FBS) and antibiotics, and buffered at pH 6.5 or 7.4 with 25 mM piperazine-N,N′-bis(2-ethanesulfonic acid) (PIPES) and 4-(2-hydroxyethyl)-1-piperazineethanesulfonic acid (HEPES). pH 6.5-adapted tumor cells were established as previously described [35,36,37]. A luciferase-expressing AB1 cell line (AB1-luc) was obtained upon infection with (firefly) luciferase-encoding lentivirus particles (Amsbio, LVP326, Abington, UK), followed by selection in 2 µg/mL puromycin. For generation of conditioned media (CM), mesothelioma cells (2 × 10^6^ cells/mL) were cultured in RPMI medium buffered at pH 6.5 or 7.4; the corresponding CM were collected after 48 h, centrifuged, and stored at −20 °C. 

### 4.2. Mice

All the experiments involving mice received the approval of the University Ethic Committee (approval ID 2016/UCL/MD018) and were carried out according to National Animal Care regulations. Balb/CByJ, C57BL/6j and OT-1 (C57BL/6-Tg (TcraTcrb)1100Mjb/Crl) mice were obtained from Charles River (Charles River Laboratories, Saint-Germain-Nuelles, France). Tumor xenografts were initiated by injecting intraperitoneally (i.p.) 1 × 10^5^ Ab-1 Luc cells in 6-week-old Balb/CByJ mice. Tumor formation and progression were evaluated by live bioluminescence imaging using PhotonIMAGER™.

### 4.3. Dendritic Cells and Vaccination

Bone marrow-derived dendritic cells (BMDCs) were generated from Balb/CByJ or C57BL/6j mice as described previously [40,61]. At day 7 post-isolation, BMDCs were either left untreated or exposed to CM treatment (dilution 1:1) for the indicated periods of time. In some experiments, DCs were exposed to 5 µM TGF-beta type I receptor inhibitor SB-431542, 15 µM DGAT1 inhibitor A922500, or 10 µM DGAT2 inhibitor PF-06424439; concentrations were chosen based on our previous work in cancer cells [36]. DC maturation was obtained by stimulation with 0.3 µg/mL LPS for 18 h and analyzed with antibodies against CD11c-BV421 (BD Pharm, 565452, San Jose, CA, USA), MHCII (I-A/I-E)-APC (BD Pharm, 565367), CD40-PE (BD Pharm, 553791), CD80-PE (eBioscience, 12-0801, San Diego, CA, USA), CD86-PE (eBioscience, 12-0862), or CCR7-PE (BioLegend 120105). Live/dead exclusion was achieved by staining with FVD eFluor780 (eBioscience, 65-0865-14). Flow cytometry analysis was performed on FACS Canto II and data were analyzed using FlowJo software (version 10.6.2). For DC vaccine generation, Ab1 cells were lysed by triple freeze–thaw processes and added to DCs at day 7. Tumor lysate-loaded DCs were treated on day 8 with 0.3 µg/mL LPS for 8 h before mouse i.p. administration (2 × 10^6^ DCs in 100 μL phosphate-buffered saline (PBS)).

### 4.4. Lipid Droplet Detection

For Oil Red O (ORO) staining, cells grown on 3 Well Chamber (Ibidi, Planeg, Germany) were fixed using 4% paraformaldehyde (PFA) (*wt/vol*) for 30 min before staining with 30 mg/mL Oil Red O solution in 60% (*vol/vol*) isopropanol for 20 min. After nuclear counterstain with hematoxylin, bright-field images were acquired at 63× magnification using an AxioImager.z1-ApoTome1 (Zeiss, Oberkochen, Germany). As a confirmation of neutral lipid staining, PFA-fixed cells were also incubated with 2 µM BODIPY 493/503 (#D3922; Thermo Fisher Scientific, Waltham, MA, USA) for 40 min at room temperature. After DAPI co-staining, slides were imaged with Cell Observer Spinning Disk confocal microscope (Zeiss, Oberkochen, Germany). ORO and BODIPY signals were analyzed using ImageJ to quantify LD-positive areas (normalized per cell nucleus).

### 4.5. TGF-β2 and IL12 Measurements

Active human TGF-β2 protein levels were assessed by using a dedicated ELISA detection kit (#DB250, R&D Systems, Minneapolis, MN, USA) as previously described [36]. Of note, TGF-β2 present in the conditioned medium was not activated by HCl treatment so that the levels of naturally active TGF-β2 could be determined. For IL12p70 measurements, a mouse IL12 detection kit protocol (BioLegend, 431411, San Diego, CA, USA) was used according to manufacturer’s instructions. 

### 4.6. Cell Death

For cell death profiling, cells were incubated with fluorescein isothiocyanate (FITC)-conjugated Annexin V (Immunostep, ANXVF-200T) and 1 μg/mL propidium iodide (PI, Sigma, St. Louis, MI, USA) according to the manufacturer’s instructions. After 15 min of incubation at room temperature in the dark, cells were analyzed by flow cytometry on FACSCanto II (BD Biosciences) with a gating strategy excluding debris and doublet cells. 

### 4.7. Metabolic Profiling

For Seahorse analysis, BMDCs were seeded (8 × 10^4^ cells/well) on Seahorse XF96 Cell Culture Microplates (Agilent, Santa Clara, CA, USA) and grown for 6 days in the presence of granulocyte macrophage-colony stimulating factor (GM-CSF) and IL-4. On day 6, medium was replaced by fresh medium (including treatments) and both the extracellular acidification rate (ECAR) and oxygen consumption rate (OCR) were analyzed in basal (non-stimulated) conditions and upon 1 µg/mL LPS injection using a Seahorse XF96 plate reader [36,62]. In some experiments, glucose consumption and lactate production were also determined using enzymatic assays (CMA 600 Analyzer, Aurora Borealis, Taiwan) [62]. 

### 4.8. In Vitro and In Vivo DC Migration 

Mature DCs (5 × 10^4^ cells in 100 µL RPMI) were added into the upper chamber of 5 µm-Transwell (Corning, Corning, NY, USA) 24-well cell culture inserts. Lower chambers were filled with RPMI containing 50 ng/mL mouse recombinant CCL21 (R&D Systems 457-6C-025). After 2 h, cells from the lower chambers were collected and stained with BV421-CD11c antibody (BD Pharm, 565452) before analysis on FACS Canto II. For in vivo DC tracking, mature DCs were labeled with VivoTrack 680 NIR Fluorescent Imaging Agent (Perkin Elmer, Waltham, MA, USA) according to the manufacturer’s recommendations. After extensive washing, labeled DCs (1 × 10^6^ in 40 µL) were injected into mouse hind footpads and at day 3, mice were euthanized and popliteal lymph nodes (PLNs) were isolated and imaged using PhotonIMAGER (Biospace Lab, Paris, France) (excitation 680 nm, emission 730 nm). Data were normalized according to the signal obtained after injection at day 0 (Biospace Lab). 

### 4.9. In Vitro T Cell Activation

DCs were plated (4 × 10^4^ cells per well) in a 48-well plate and treated as indicated on day 6. DCs were activated for 18 h using 0.3 µg/mL LPS and 500 ng/mL OVA peptide (SIINFEKL) (Iba Solutions, 6-7015-901, Dakula, GA, USA). After washing, isolated CD8^+^ OT-1 were added (ratio DC/T cell 1:5) and T cell response was analyzed after 48 h. Proliferation of CD3^+^/CD8^+^ T cells was analyzed based on dilution of CFSE while for measuring activation, T cells were treated for 4 h with Golgi Stop (BD Pharm, 554715) before exposure to Fixation/Permeabilization Solution (BD Pharm 554715) and staining with phycoerythrin-conjugated interferon gamma (IFNγ-PE) antibody (BioLegend, 505807). Flow cytometry analysis was performed on FACS Canto II (BD Biosciences) using CD3-APC (BD Pharm, 553066), CD8a-FITC (BD Pharm, 553031), and CD69-BV421 (BD Pharm, 562920); data were analyzed using FlowJo software. Live/dead exclusion was achieved by staining with FVD eFluor780 (eBioscience, 65-0865-14).

### 4.10. Ex Vivo T Cell Activation

Balb/CByJ mice were euthanized and spleens were surgically removed and placed in PBS. Single-cell solution was obtained by passing an organ through 70-µm strainer. Splenocytes were separated by gradient centrifugation with Ficoll-Paque™ PLUS (VWR). Red blood cells were removed using RBC lysis buffer (eBioscience, 00-4333-57). Splenocytes were cryopreserved in FBS with 10% DMSO until functional experiments. For activation, splenocytes were co-cultured with Ab1 cells at a ratio of 10:1. 

Staining was performed using IFNγ-PE antibody (BioLegend, 505807) and CD107-PE (LAMP-1) antibody (BioLegend, 121611) together with protein transport inhibition with Golgi Stop (BD Pharm, 554715). T cell proliferation was determined based on dilution of CellTrace™ CFSE dye (Thermo Fisher). CD4+ activation was analyzed using CD25-PE antibody (BioLegend, 102007) and CD134 (OX40)-BV421 antibody (BioLegend, 119411).

### 4.11. Statistics

Data were expressed as mean ± SEM of at least three independent experiments. Statistical significance between experimental conditions was determined by Student’s t-test or one-way analysis of variance (ANOVA, Tuckey’s post-hoc test). All data were analyzed with GraphPad Prism 7.0 (San Diego, CA, USA).

## 5. Conclusions

In conclusion, our study identified that the acidic TME is associated with DC dysfunction through pharmacologically reversible TGF-β2-dependent mechanisms including (1) LD accumulation and a profound rewiring of metabolic preferences that account for a dramatic defect in CCR7-dependent migration, (2) decreases in the expression of co-stimulatory CD40 molecule and in the secretion of pro-inflammatory interleukin-12, and (3) a consecutive reduction in Ag-specific CD8^+^ expansion together with a lesser-associated IFN-γ signaling. These findings open new perspectives to fine-tune immunotherapeutic approaches in order to target cold tumors such as MM.

## Figures and Tables

**Figure 1 cancers-12-01284-f001:**
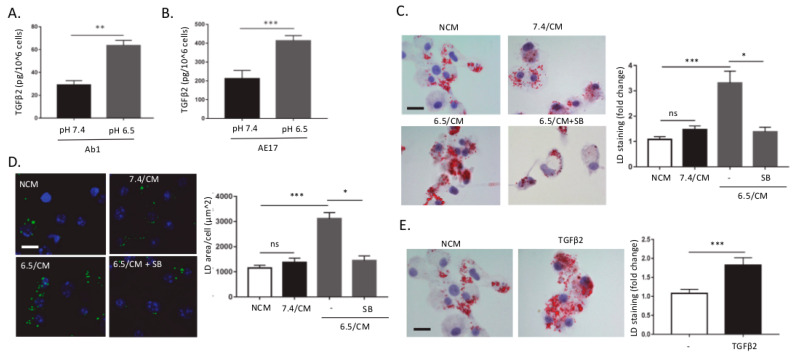
TGF-β2-dependent lipid droplet accumulation in dendritic cells in response to the acidic mesothelioma milieu. (**A**,**B**) Control (pH 7.4) and acidosis (pH 6.5)-adapted Ab1 (**A**) and AE17 (**B**) mesothelioma cells were grown for 48 h, and active TGF-β2 secretion was assayed using ELISA. (**C**–**E**) Dendritic cells (DCs) were incubated with non-conditioned medium (NCM) or treated for two days either with conditioned medium (CM) from mesothelioma cells maintained at pH 7.4 or pH 6.5 (7.4/CM and 6.5/CM, respectively) (**C**,**D**) or with 4 ng/mL recombinant TGF-β2 (**E**). In some experiments, DCs were also exposed to 5 µM SB-431542. Representative pictures of lipid droplet (LD) content as determined using Oil Red O (ORO) (scale = 20 µm) (**C**,**E**) or BODIPY 495/503 staining (scale: 20 µm, green: BODIPY 495/503, blue: DAPI) (**D**) are shown together with quantification of the cellular area covered by LDs (*n* = 3, * *p* < 0.05, ** *p* < 0.01, *** *p* < 0.001; ns = non-significant).

**Figure 2 cancers-12-01284-f002:**
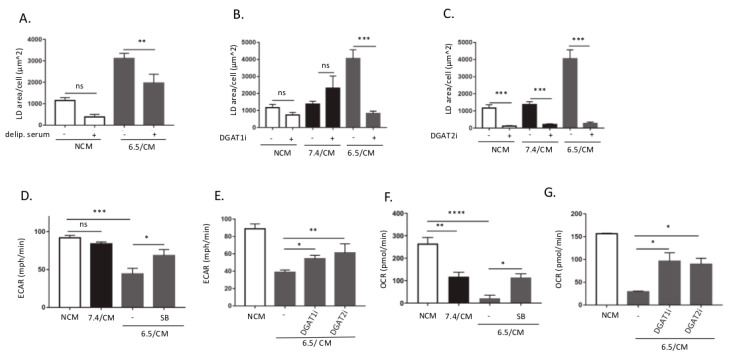
Diacylglycerol O-acyltransferase (DGAT)-dependent LD accumulation in dendritic cells (DCs) leads to metabolic reprogramming. DCs were incubated with non-conditioned medium (NCM) or treated for 2 or 3 days either with conditioned medium from AE17 or Ab1 mesothelioma cells maintained at pH 7.4 or pH 6.5 (7.4/CM and 6.5/CM, respectively). (**A**–**C**) Effects of 6.5/CM with or without delipidated serum (**A**), 15 µM A922500 (DGAT1i) (**B**), or 10 µM PF-06424439 (DGAT2i) (**C**) on cellular LD content, as determined using BODIPY 495/503 (*n* = 3, ** *p* < 0.01, *** *p* < 0.001; ns = non-significant). (**D**–**G**) Effects of 6.5/CM with or without 5 µM SB-431542 and either DGAT1i or DGAT2i on the extracellular acidification rate (ECAR) (**D**,**E**) and oxygen consumption rate (OCR) (**F**,**G**), as detected using the Seahorse XF Analyzer (*n* = 3, * *p* < 0.05, ** *p* < 0.01, *** *p* < 0.001, **** *p* < 0.0001; ns = non-significant).

**Figure 3 cancers-12-01284-f003:**
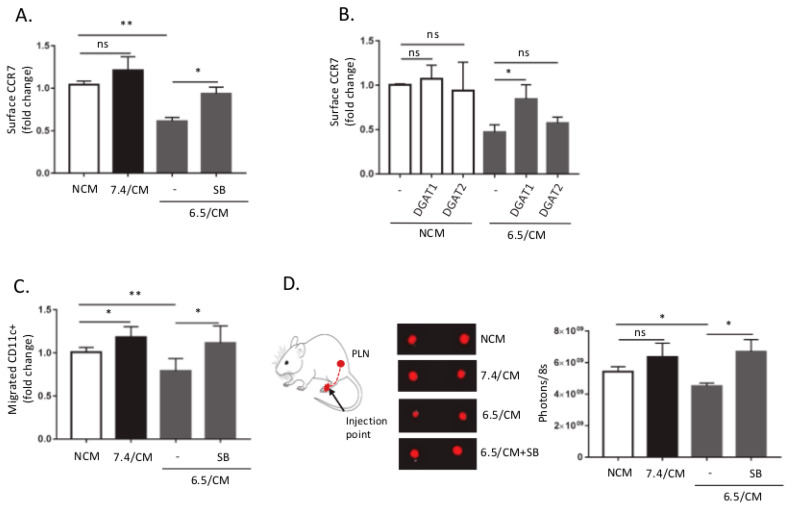
Dendritic cells (DCs) exposed to acidic mesothelioma milieu exhibit reduced migratory potential in vitro and in vivo. DCs were incubated with non-conditioned medium (NCM) or treated for 2 days either with conditioned medium from Ab1 mesothelioma cells maintained at pH 7.4 or pH 6.5 (7.4/CM and 6.5/CM, respectively). (**A**,**B**) Effects of 6.5/CM with or without 5 µM SB-431542 (**A**) and either 15 µM A922500 (DGAT1i) or 10 µM PF-06424439 (DGAT2i) (**B**) on CCR7 surface expression as determined by flow cytometry (*n* = 3, * *p* < 0.05, ** *p* < 0.01, ns = non-significant). (**C**,**D**) Effects of 5 µM SB-431542 on DC migration determined either in vitro in transwell migration assay (*n* = 3, * *p* < 0.05, ** *p* < 0.01, ns = non-significant) (**C**) and in vivo in the mouse footpad assay as depicted in (**D**) together with representative pictures and quantification of VivoTrack 680 NIR fluorescence signal into popliteal lymph nodes collected 3 days post-injection (*n* = 5−6, * *p* < 0.05; ns = non-significant).

**Figure 4 cancers-12-01284-f004:**
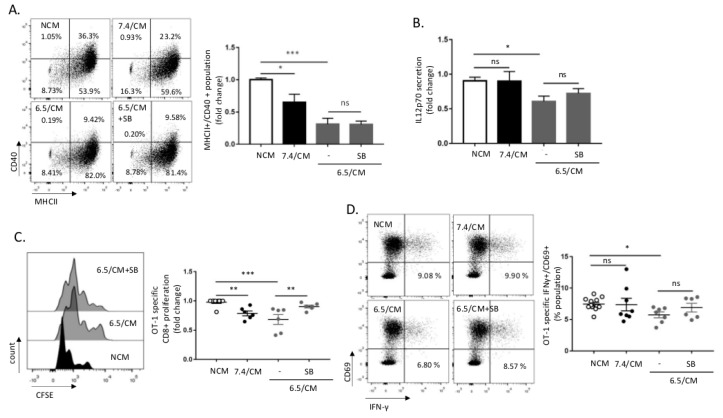
CD8^+^ T cell response is altered upon dendritic cells (DCs) exposure to the mesothelioma acidic milieu. DCs were incubated with non-conditioned medium (NCM) or treated for 2 days either with conditioned medium from AE17 or Ab1 mesothelioma cells maintained at pH 7.4 or pH 6.5 (7.4/CM and 6.5/CM, respectively). (**A**,**B**) Effects of 6.5/CM with or without 5 µM SB-431542 on MHCII^+^/CD40^+^ surface expression as determined by flow cytometry (**A**) and pro-inflammatory interleukin-12 (IL-12p70) secretion as detected by ELISA (**B**). (**C**,**D**) Effects of 6.5/CM with or without 5 µM SB-431542 on OT-1 specific CD8^+^ proliferation, as measured using carboxyfluorescein succinimidyl ester (CFSE) dilution (**C**) and activation, as evaluated by determining CD69^+^/IFN-γ^+^ frequency (**D**). The charts and histogram presented in this figure are representative of at least three different experiments. * *p* < 0.05, ** *p* < 0.01, *** *p* < 0.001; ns = non-significant.

**Figure 5 cancers-12-01284-f005:**
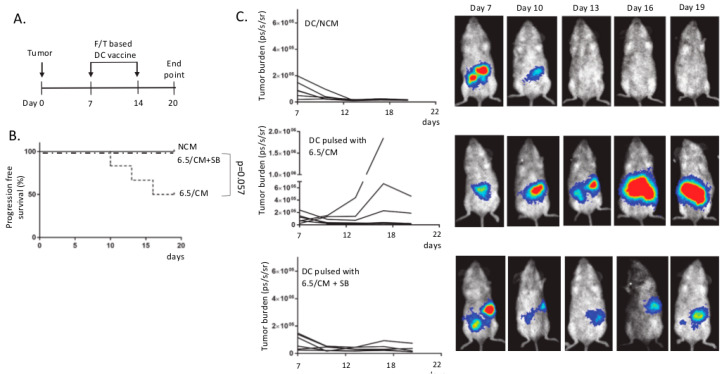
Acidic mesothelioma milieu alters the therapeutic efficacy of dendritic cell (DC)-based vaccination. Balb/c mice were injected intraperitoneally (i.p.) with 1 × 10^5^ Ab1-luc and were vaccinated at day 7 and 14 with DCs pulsed with mesothelioma cell lysates in the presence of either non-conditioned medium (NCM), 6.5/CM, or 6.5/CM + 5 µM SB-431542. (**A**) Protocol of autologous DC vaccination regimen. (**B**) Kaplan–Meier curves depicting progression-free survival for each condition (*n* = 6 mice per group). (**C**) Time course of peritoneal mesothelioma growth (*n* = 6 mice per group) and representative pictures of bioluminescence signal measurements.

**Figure 6 cancers-12-01284-f006:**
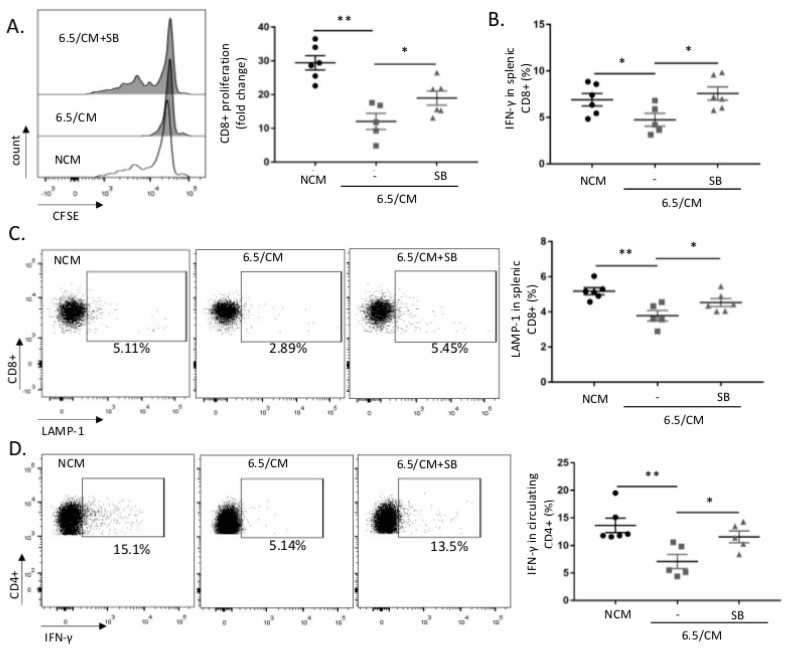
The acidic mesothelioma milieu alters the ability of dendritic cell (DC)-based vaccine to induce anti-tumor T cell activation in vivo. (**A**–**C**) CD8^+^ splenocytes collected at day 14 post-vaccination (i.e., NCM, 6.5/CM, and 6.5/CM + 5 µM SB-431542, see Figure 5) were re-exposed to Ab1 mesothelioma cells. Proliferation of CD8^+^ splenocytes was detected based on CFSE dilution (*n* = 6, * *p* < 0.05, ** *p* < 0.01) (**A**) and activation was probed based on interferon gamma (IFN-γ) production (**B**) and lysosomal-associated membrane protein-1 (LAMP-1) staining (**C**) (*n* = 6, * *p* < 0.05, ** *p* < 0.01). (**D**) IFN-γ production in circulating CD4^+^ lymphocytes stimulated with phorbol myristate acetate (PMA)/ionomycin (*n* = 6, * *p* < 0.05, ** *p* < 0.01).

**Figure 7 cancers-12-01284-f007:**
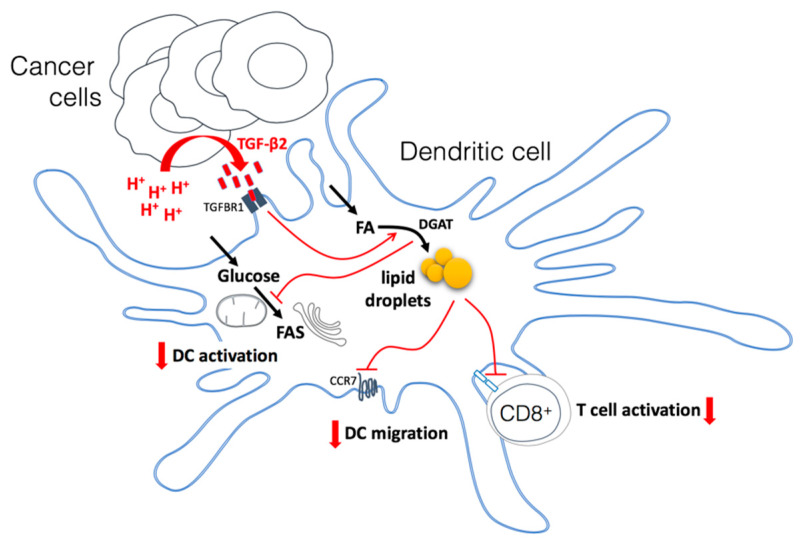
Model depicting how the acidic tumor microenvironment promotes TGF-β2 secretion by cancer cells which in turn leads to lipid droplet (LD) accumulation and profound metabolic rewiring in dendritic cells (DCs). DCs exposed to a TGF-β2-containing mesothelioma acidic milieu exhibit a net increase in the uptake of fatty acids that accumulate as triglycerides into LDs, as well as a dramatic reduction in glucose metabolism. The consequences are a deficit in both DC migratory potential and activation as well as a decrease in T cell activation. Inhibition of TGF-β2 signaling but also diacylglycerol O-acyltransferase (DGAT), the last enzyme involved in triglyceride synthesis, has the potential to restore DC activity and anticancer immune response. TGFBR1: TGF-beta type I receptor inhibitor; CCR7: C-C chemokine receptor type 7; FAS: fatty acid synthesis.

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
