# Peer review of "Acidosis-Induced TGF-β2 Production Promotes Lipid Droplet Formation in Dendritic Cells and Alters Their Potential to Support Anti-Mesothelioma T Cell Response"

_cancers, 2020, doi:10.3390/cancers12051284_

Round 1

Reviewer 1 Report

Natalia Trempolec et al presented new evidence of one of tumor microenvironment (TME) factor acidosis mediated lipid accumulation in dendritic cells (DC)  that alters the crosstalk between DC and cytotoxic T cells. The mechanism underlying this is that acidosis induced increased autocrine  TGF-β2 secretion by mesothelioma cancer cells, which is a key factor for the communication between cancer cells and DC. Furthermore, they showed that inhibition of TGF-β2 signaling and triglyceride synthesis inhibitor diacylglycerol O-acyltransferase induced a significant restoration of DC activity and anticancer immune response. This study is a continuation of their previous studies and provides guidance for the potential clinical application for the mesothelioma immunotherapy.  Some comments:

  1. Fig1E, SB-431542 treatment alone and SB/TGFB2 should be included as controls.
  2. Fig2A (NCM delip. vs 6.5/CM delip.) shows that uptake of exogenous lipis plays only a minor part in the reduction of LD accumulation. How does the uptake of amino acids and the glucose alter during the 6.5/CM treatment should be investigated?  FigS2 suggests that TGFB2 mediated LD accumulation is not mediated by increased glucose uptake. The gene expression of enzymes related to lipid uptake, lipid oxidation, and lipid secretion, de novo lipogenesis or other methods should also be investigated to understand the source of LD accumulation. Are the lipid-related transcription factors SREBP1, ChREBP1 activated?
  3. Fig3. how to explain that DGAT2i did not inhibit DC migrational activity, but inhibit the DC lipid accumulation the same as DGAT1i. It seems to suggest that the lipid accumulation is independent of the migrational activity as well as antigen presentation showing in Fig4, as the inhibitors (DGATi, SB) differentially affect the various aspects of DC, which are the key conclusion of this manuscript.
  4.  A graph should be provided to show the conclusion of this work.

Author Response

REVIEWER 1

Natalia Trempolec et al presented new evidence of one of tumor microenvironment (TME) factor acidosis mediated lipid accumulation in dendritic cells (DC)  that alters the crosstalk between DC and cytotoxic T cells. The mechanism underlying this is that acidosis induced increased autocrine TGF-β2 secretion by mesothelioma cancer cells, which is a key factor for the communication between cancer cells and DC. Furthermore, they showed that inhibition of TGF-β2 signaling and triglyceride synthesis inhibitor diacylglycerol O-acyltransferase induced a significant restoration of DC activity and anticancer immune response. This study is a continuation of their previous studies and provides guidance for the potential clinical application for the mesothelioma immunotherapy.  Some comments:

  1. Fig1E, SB-431542 treatment alone and SB/TGFB2 should be included as controls.

We know refer to these controls in the text; SB431542 completely blunted TGF-β2 effects and did not show significant effects when used alone

  1. Fig2A (NCM delip. vs 6.5/CM delip.) shows that uptake of exogenous lipids plays only a minor part in the reduction of LD accumulation. How does the uptake of amino acids and the glucose alter during the 6.5/CM treatment should be investigated?  FigS2 suggests that TGFB2 mediated LD accumulation is not mediated by increased glucose uptake. The gene expression of enzymes related to lipid uptake, lipid oxidation, and lipid secretion, de novo lipogenesis or other methods should also be investigated to understand the source of LD accumulation. Are the lipid-related transcription factors SREBP1, ChREBP1 activated?

It is true that in Figure 2 the extent of LD in the presence of delipidated serum is not null. Still it should be noted that at the beginning of the experiment DC already contained a certain amount of LD as documented in the control condition (i.e., Fig. 2A, NCM, without delipidated serum). If this LD level in control condition is considered as basal, the extent of the reduction in LD accumulation in 6.5/CM delip. condition actually reached 60%. This justified our words: “this net reduction in LD formation suggests that accumulation of LD was largely dependent on the uptake of exogenous lipids”. As emphasized by the Reviewer, we also showed that DC exposed to mesothelioma acidic milieu exhibit both a decrease in both EACR (i.e. glycolysis) and OCR reflecting a lesser TCA cycle activity. This accounts for a defect in production of NADPH, glycerol and citrate as a source of acetylCoA, further supporting a reduced FA synthesis. Finally, the differential effect of DGAT1 and DGAT2 inhibitors may also account for a preferred exogenous source of FA (vs. de novosynthesis) (see below). Exploring the transcriptome related to lipid metabolism would certainly bring new insights but at this stage we believe our work per sesupports a major role of FA uptake in LD accumulation under acidosis/TGF-b2 exposure. To abide by the Reviewer’s comment, we have however modified the text to read: “Although we cannot formally exclude a contribution of FA synthesis to LD formation, …”.

  1. how to explain that DGAT2i did not inhibit DC migrational activity, but inhibit the DC lipid accumulation the same as DGAT1i. It seems to suggest that the lipid accumulation is independent of the migrational activity as well as antigen presentation showing in Fig4, as the inhibitors (DGATi, SB) differentially affect the various aspects of DC, which are the key conclusion of this manuscript.

Longstanding questions about DGAT1 and DGAT2 are how and why two fundamentally different enzymes, that are evolutionarily unrelated, contribute to TG storage. The current model is that DGAT2 has a more ancient function for mediating TG synthesis of de novosynthesized fatty acids while DGAT1 would be more important in protecting the ER from lipotoxic effects of excess exogenous fatty acids (PMID: 18757836, 28768178). Whether this specificity explains why inhibiting DGAT2 poorly influenced CCR7 expression while DGAT1 inhibition fully restored its expression (Figure 3B) is a possibility. Moreover, our observations are that DGAT1 inhibitor blunted LD accumulation in 6.5/CM-treated cells but not in 7.4/CM-treated cells (Fig. 2B) while DGAT2 indistinctly inhibited LD formation in both conditions (Fig. 2C). Altogether, this may be interpreted as a more specific role of DGAT1 in regulating LD accumulation in response to TGF-b2, indirectly supporting that LD formation is a way for DCunder acidosis to handle the uptake of excess exogenous (and not de novo) fatty acids. We have now mentioned the information about the different roles of DGAT1 and DGAT2 in the discussion of our manuscript.

  1. A graph should be provided to show the conclusion of this work.

We thank the Reviewer for this suggestion. A graph summarizing our data and depicting our model is now presented as Figure 7.

Reviewer 2 Report

In the manuscript submitted by Trempolec et al, the authors describe how acidosis leads to TGF-beta secretion by mesothelioma cells which then causes metabolic rewiring in dendritic cells, thereby altering anti-cancer immune response. Overall, I find the manuscript nicely written in good and proficient English. Considering the content, I have only minor suggestions for the manuscript before considering publication in Cancers

  • Throughout the manuscript, the inhibitors SB-431542, A922500 and PF-06424439 are used for in vitro studies. However, the inhibitor treatment does not appear in the Material and Methods section. Could the authors comment on how the concentrations used in these experiments were determined? Are there any functional readouts? Specifically when the authors postulate that only DGAT2 inhibition causes basal amounts of LD (Fig. 2C( how can they be sure this is not just because of wrong inhibitor concentrations? Can the same effect be achieved by knockdown of DGAT1 and DGAT2? 
  • In Fig 1D, the figure legend does not explain green and blue staining
  • In Figure 5, could the authors comment on the remaining bioluminiscent signal present after 19 days in the CM+SCB group?
  • Can the authors comment more on the different effects of DGAT1 and DGAT2 inhibition? Is there a mechanistical explanation why DGAT1 inhibition works differently on CCR7 expression?

Author Response

REVIEWER 2

In the manuscript submitted by Trempolec et al, the authors describe how acidosis leads to TGF-beta secretion by mesothelioma cells which then causes metabolic rewiring in dendritic cells, thereby altering anti-cancer immune response. Overall, I find the manuscript nicely written in good and proficient English. Considering the content, I have only minor suggestions for the manuscript before considering publication in Cancers.

Throughout the manuscript, the inhibitors SB-431542, A922500 and PF-06424439 are used for in vitro studies. However, the inhibitor treatment does not appear in the Material and Methods section. Could the authors comment on how the concentrations used in these experiments were determined? Are there any functional readouts? Specifically when the authors postulate that only DGAT2 inhibition causes basal amounts of LD (Fig. 2)how can they be sure this is not just because of wrong inhibitor concentrations? Can the same effect be achieved by knockdown of DGAT1 and DGAT2? 

Information related to the inhibitors and respective dosage is now provided in the Materials and Methods section as well as in the Figure legends. As indicated, concentrations were chosen based on our previous work (ref. #36) where inhibition of phospho-Smad signaling cascade was documented with 5 µM TGF-beta type I receptor inhibitor SB-431542 and inhibition of LD accumulation was obtained with 15 µM DGAT1 inhibitor A922500. These dosages as well as the 10 µM concentration for DGAT2 inhibitor PF-06424439 were derived from the literature where their inhibitory potential was originally documented. We agree with the Reviewer that we cannot exclude that higher concentrations could have led to more profound effects in Figure 2C but also in Figure 3B.  As emphasized below, the issues related to the differential effects of DGAT1 vs.DGAT2 inhibition go beyond the concentrations of the inhibitors and further work is needed to address this important question. As suggested by the Reviewer, this will require specific knock-down cell modelsand even knock-out mice instead of pharmacological inhibitors, the concentration and the kinetics of which influencingthe read-outs.This limitation is now indicated in the text.

In Fig 1D, the figure legend does not explain green and blue staining

The Figure legend was modified accordingly (green - BODIPY 495/503, blue – DAPI).

In Figure 5, could the authors comment on the remaining bioluminescent signal present after 19 days in the CM+SCB group?

We could have chosen pictures where bioluminescent signals had vanishedafter 2 weeks but as shown in the associated graph, some mice had still some background bioluminescent signals after 19 days. This signal indicates that some viable cancer cells are still present in mice. Whether these cells could in fineescape the vaccination protocol and leads to tumor relapse is unknown as well as whether a third vaccine administration could have definitely led to tumor eradication. Still, we believe this Figure (as well Figure S5) supports the critical role of TGF-b2 present in the acidic milieu of mesothelioma as a cause of DC dysfunction.

Can the authors comment more on the different effects of DGAT1 and DGAT2 inhibition? Is there a mechanistical explanation why DGAT1 inhibition works differently on CCR7 expression?

Longstanding questions about DGAT1 and DGAT2 are how and why two fundamentally different enzymes, that are evolutionarily unrelated, contribute to TG storage. The current model is that DGAT2 has a more ancient function for mediating TG synthesis of de novosynthesized fatty acids while DGAT1 would be more important in protecting the ER from lipotoxic effects of excess exogenous fatty acids (PMID: 18757836, 28768178). Whether this specificity explains why inhibiting DGAT2 poorly influenced CCR7 expression while DGAT1 inhibition fully restored its expression (Figure 3B) is a possibility. Moreover, our observations are that DGAT1 inhibitor blunted LD accumulation in 6.5/CM-treated cells but not in 7.4/CM-treated cells (Fig. 2B) while DGAT2 indistinctly inhibited LD formation in both conditions (Fig. 2C). Altogether, this may be interpreted as a more specific role of DGAT1 in regulating LD accumulation in response to TGF-b2, indirectly supporting that LD formation is a way for DCunder acidosis to handle the uptake of excess exogenous (and not de novo) fatty acids. We have now mentioned the information about the different roles of DGAT1 and DGAT2 in the discussion of our manuscript.

Reviewer 3 Report

The article entitled, Acidosis-Induced TGF-β2 Production Promotes Lipid Droplet Formation in Dendritic Cells and Alters Their Potential to Support Anti-Mesothelioma T cell Response, addresses a relevant topic and had well-designed methods. In my opinion, the manuscript can be considered for publication. However, minor corrections must be made:

- Please review the text captions as the β and γ symbols.
- Besides, the first time you use the acronym, inform what it refers to.
- INF-γ or IFN-γ

Author Response

REVIEWER 3

The article entitled, Acidosis-Induced TGF-β2 Production Promotes Lipid Droplet Formation in Dendritic Cells and Alters Their Potential to Support Anti-Mesothelioma T cell Response, addresses a relevant topic and had well-designed methods. In my opinion, the manuscript can be considered for publication. However, minor corrections must be made:

- Please review the text captions as the β and γ symbols.

The text was modified accordingly.

- Besides, the first time you use the acronym, inform what it refers to.

The text was modified accordingly.

- INF-γor IFN-γ

The text was modified accordingly to read IFN-γ.

Reviewer 4 Report

This study propose that acidic mesothelioma drives DC dysfunction and altered T cell response through pharmacologically reversible TGF-β2-dependent mechanism. The study was interested; however, several important information was missing. Please carefully rechecked and revised.

EMT mentioned in results 2.1 should be added with full name.

Many formats of TGF-β2 and IFN-γ are incorrect, please check full manuscript.

Please mentioned the repeated time of each experiment.

Please provide full name of PIPES and HEPES.

The cat number of luciferase-encoding lentivirus should be mentioned in materials and methods.

The version of Flowjo should be added.

Please add space between number and unit. For example, page 11 ‘’5 mm’’.

Mice species should be OT-1 but not OT-I (page 12), please check full manuscript.

Which species of mice was used for T cell activating experiment?

Author Response

REVIEWER 4

This study propose that acidic mesothelioma drives DC dysfunction and altered T cell response through pharmacologically reversible TGF-β2-dependent mechanism. The study was interesting; however, some information is missing. Please carefully rechecked and revised.

 EMT mentioned in results 2.1 should be added with full name.

The text was modified to mention epithelial-to-mesenchymal transition (EMT).

Many formats of TGF-β2 and IFN-γare incorrect, please check full manuscript.

The text was edited accordingly.

Please mentioned the repeated time of each experiment.

We made sure this information was available in the Figure legends of our revised manuscript.

Please provide full name of PIPES and HEPES.

The text was modified to mention piperazine-N,N′-bis(2-ethanesulfonic acid) (PIPES) and 4-(2-hydroxyethyl)-1-piperazineethanesulfonic acid (HEPES).

The cat number of luciferase-encoding lentivirus should be mentioned in materials and methods.

The text was modified to mention #LVP326.

The version of Flowjo should be added.

The text was modified to mention version 10.6.2.

Please add space between number and unit. For example, page 11 ‘’5 mm’’.

The text was modified accordingly.

Mice species should be OT-1 but not OT-I (page 12), please check full manuscript.

The text was modified accordingly.

Which species of mice was used for T cell activating experiment?

The text was modified to mention Balb/CByJ mice.